# Mechanisms of PD-L1 Regulation in Malignant and Virus-Infected Cells

**DOI:** 10.3390/ijms22094893

**Published:** 2021-05-05

**Authors:** Hadia Farrukh, Nader El-Sayes, Karen Mossman

**Affiliations:** 1School of Interdisciplinary Science, Faculty of Science, McMaster University, Hamilton, ON L8S 4K1, Canada; farrukhh@mcmaster.ca; 2Department of Biochemistry and Biomedical Sciences, McMaster Immunology Research Centre, McMaster University, Hamilton, ON L8S 4K1, Canada; elsayesn@mcmaster.ca; 3Department of Medicine, McMaster Immunology Research Centre, McMaster University, Hamilton, ON L8S 4K1, Canada

**Keywords:** cancer immunotherapy, immune checkpoint blockade, PD-1, PD-L1, combination therapy

## Abstract

Programmed cell death protein 1 (PD-1), a receptor on T cells, and its ligand, PD-L1, have been a topic of much interest in cancer research. Both tumour and virus-infected cells can upregulate PD-L1 to suppress cytotoxic T-cell killing. Research on the PD-1/PD-L1 axis has led to the development of anti-PD-1/PD-L1 immune checkpoint blockades (ICBs) as promising cancer therapies. Although effective in some cancer patients, for many, this form of treatment is ineffective due to a lack of immunogenicity in the tumour microenvironment (TME). Despite the development of therapies targeting the PD-1/PD-L1 axis, the mechanisms and pathways through which these proteins are regulated are not completely understood. In this review, we discuss the latest research on molecules of inflammation and innate immunity that regulate PD-L1 expression, how its expression is regulated during viral infection, and how it is modulated by different cancer therapies. We also highlight existing research on the development of different combination therapies with anti-PD-1/PD-L1 antibodies. This information can be used to develop better cancer immunotherapies that take into consideration the pathways involved in the PD-1/PD-L1 axis, so these molecules do not reduce their efficacy, which is currently seen with some cancer therapies. This review will also assist in understanding how the TME changes during treatment, which will provide further rationale for combination therapies.

## 1. Introduction

Programmed cell death protein 1 (PD-1) (also known as CD279) is a co-inhibitory receptor on T cells that, when bound to its ligand PD-L1, inhibits T cell activation and cytotoxicity [1]. One of the important outcomes of this normal interaction is the prevention of autoimmunity by suppressing proliferation of self-reactive T cells [2]. PD-1 is expressed by a variety of innate and adaptive immune cells including natural killer (NK) cells, monocytes, dendritic cells (DCs), NKT cells, T cells, and B cells [1]. One of the ligands of PD-1 is PD-L1 (also known as CD274 and B7-H1), which is expressed by all hematopoietic cells, many nonhematopoietic cells (such as endothelial, epithelial, and mesenchymal stem cells), as well as many tumour cells [1]. In contrast, the other ligand, PD-L2 (also known as CD273 and B7-DC), is only expressed by non-hematopoietic cells and antigen presenting cells (APCs) [1]. However, recent studies have shown that PD-L2 may also be expressed in some tumours [2,3]. Upregulation of PD-1 on T cells and increased expression of PD-L1 have been associated with T cell exhaustion, which occurs when these immune cells are repeatedly stimulated by antigens, or in this case, by strong binding of PD-L1 to its receptor [4]. The exhausted T cell phenotype is characterized by less proliferation, loss of effector functions, and apoptosis of the T cell, which is a normal part of the immune response to prevent excess tissue damage due to inflammation [4].

Viruses have adapted to human host responses to achieve successful viral infection through host immune evasion. Some ways in which this is achieved is by inhibiting interferon production, degradation of specific receptors to prevent signaling of the immune system, and by shutting off host protein synthesis [5]. Another way viruses evade the immune system is through upregulation of PD-L1 leading to inhibition of T cell activation [1]. However, PD-L1 regulation during viral infection is complex and differs depending on whether the host has an acute virus infection, such as with lymphocytic choriomeningitis virus (LCMV), or chronic virus infection, such as with human immunodeficiency virus (HIV) [4]. More research is required on how the differences in PD-L1 expression are manifested in a variety of infections. 

Some tumour cells also upregulate PD-L1 expression to suppress cytotoxic T cells and enhance tumour survival [6]. In these highly immunogenic “hot” tumours, where there is infiltration and subsequent silencing of T cells by tumour cells, anti-PD-1/PD-L1 immune checkpoint blockades (ICBs) have been highly effective therapies [6]. However, these drugs, which include pembrolizumab, nivolumab, atezolizumab, durvalumab and avelumab, are ineffective in patients with “cold” tumours that lack T cells and other immune cells in the TME [6]. Since viruses upregulate PD-L1, oncolytic viruses have been used in conjunction with ICBs as one way to attract the immune system to the tumour and increase T cell activity [6]. 

Despite the development of treatments that block interactions between PD-1 and its ligands, regulation of PD-L1 is not completely understood. Tumours and viruses both induce host immune responses that can lead to upregulation of PD-L1 in the process of restoring homeostasis and eliminating foreign particles in the host [1,4]. Therefore, it is possible that molecules involved in inflammation and immunity may regulate the expression of PD-L1. The purpose of this review is to discuss which molecules involved in inflammation and immunity regulate PD-L1 expression and through which pathways. We also summarize the latest research on how different cancer therapies modulate PD-L1 expression and those that are being researched for combination therapies with anti-PD-1/PD-L1 ICBs. Understanding the pathways through which PD-L1 expression is regulated will help researchers improve cancer therapies by preventing upregulation of PD-L1 during treatment and the subsequent suppression of T cell effector functions. It will also lead to more research and improvement on combination therapies for cancer.

## 2. The PD-1/PD-L1 Axis and its Role in Autoimmunity and Cancer

PD-1 is a transmembrane receptor that has two tyrosine residues in its cytoplasmic domain, an immunoreceptor tyrosine-based inhibitory motif (ITIM) and an immunoreceptor tyrosine-based switch motif (ITSM) [4,7,8,9]. The ligand, PD-L2, has greater affinity to PD-1 but PD-L1 is expressed by a greater variety of cells, and therefore, this review will mostly discuss the relationship between PD-L1 and PD-1 [4]. When PD-1 ligation occurs, with PD-L1 or PD-L2, the cytoplasmic domain is phosphorylated leading to recruitment of the protein tyrosine phosphatase, SHP-2, to ITSM, which is followed by dephosphorylation of CD-3ζ and ZAP70, molecules associated with the T cell receptor (TCR) (Figure 1) [4,7,8,9]. SHP-2 also dephosphorylates the T cell costimulatory receptor, CD28, which assists with inhibition of the T cell [4,7,8,9]. Downstream signaling is also disrupted after this, leading to the inhibition of T cell activation, proliferation, differentiation, and effector functions including cytokine production, as well as the development of CD4^+^ T cells into Foxp3^+^ regulatory T cells (Tregs) [7].

Under normal conditions, PD-1 is upregulated when T cells are activated and it mostly regulates late immune responses that take place in peripheral tissues [4,7]. The main pathway to PD-L1 production is through secretion of the cytokine, interferon gamma (IFN-γ), by T cells, which activates the JAK-STAT pathway, leading to binding of the transcription factor, IRF1, to the PD-L1 promoter [10]. PD-L2 expression is mainly influenced by IFN-γ and IFN-β, which lead to the transcription factors, STAT3 and IRF1, to bind to the PD-L2 promoter [4]. The natural functions of the PD-1/PD-L1 interaction in the body are regulation, prevention of tissue damage, and prevention of T-cell cytotoxicity, all of which, if imbalanced, can lead to autoimmunity. Tumour cells take advantage of this axis to enhance growth and evade the immune system, specifically T cells.

The PD-1/PD-L1 axis plays a role in maintaining tolerance. A disruption in this homeostasis can lead to autoimmunity, where T cells no longer recognize certain antigens as self and attack the body’s tissues [11]. Many autoimmune diseases have been found to be associated with the disruption of the PD-1/PD-L1 axis, including rheumatoid arthritis (RA), systemic lupus erythematosus (SLE), and multiple sclerosis (MS) [11,12]. It should be expected that PD-L1 expression is low in autoimmune conditions. However, many of these diseases are associated with increased PD-L1 levels [8]. It is reasoned that PD-L1 and PD-1 expression induced in these autoimmune diseases may possibly be to reduce autoimmune effects [8,13,14]. Additionally, studies have shown that presence of T cells that are absent or low in PD-1 expression may contribute to autoimmune diseases, indicating that even if PD-L1 is being expressed, T cells are not inactivated by them due to low or absent PD-1 expression [8]. Certain single nucleotide polymorphisms (SNPs) in the *PDCD1* gene, which codes for PD-1, are also associated with a variety of autoimmune disorders, including RA, SLE, diabetes mellitus, and MS, in different ethnic groups [8]. Presence of these SNPs can be used to identify the risk of developing different autoimmune conditions in individuals of certain ethnicities. Subsequently, precautions can be taken to reduce risks by introducing changes in lifestyle choices, for example. Taken together, more research is required when developing therapeutics for autoimmunity if the PD-1/PD-L1 axis is targeted.

The role of the PD-1/PD-L1 axis is perhaps most extensively studied in the context of cancer, leading to the development of anti-PD-1/PD-L1 ICBs. Cancer cells express PD-L1 constitutively due to gene amplification or due to activation of oncogenic pathways, to inhibit immune responses against the tumour—this is known as innate immune resistance [15]. There is also adaptive immune resistance, where tumour and immune cells express PD-L1 in response to inflammatory mediators produced by different cells in the TME [15]. PD-L1 expression is upregulated in various types of cancers and it was found to correlate with lower overall survival and disease-free survival in human malignant tumours, suggesting that it may be one of the ways to predict clinical outcomes after anti-PD-1/PD-L1 therapy [16]. PD-L1 can also directly deliver intracellular anti-apoptotic signals to tumour cells without interaction with its receptor, further assisting tumour cells in surviving cytotoxicity [10]. The use of anti-PD-1/PD-L1 blockades as treatment of cancer has yielded variable responses depending on cancer type and this will be discussed in detail in the last section of this review.

The localization of PD-L1 expression on different cell types in the TME is important to consider. Many studies have shown that PD-L1 expression on cancer cells can inhibit cytotoxic T cell killing in vitro and inhibit antitumour immunity in vivo [17,18,19]. Furthermore, knockdown or knockout of PD-L1 in cancer cells sensitizes cancer cells to in vitro T cell killing and improves antitumour responses in murine models [17,18,20,21]. On the other hand, several studies claim that PD-L1 expression on immune cells and not cancer cells is a marker of favorable prognosis in cancer patients. One study suggests that PD-L1 expression on immune cells and not tumour cells is associated with favorable prognostic outcomes in patients with head and neck squamous cell cancer (HNSCC) [22]. A similar study found that in 500 non–small cell lung cancer (NSCLC) patients, PD-L1 expression on CD68^+^ macrophages was associated with increased CD8^+^ T cell infiltration and better response to anti-PD-1/PD-L1 therapy [23]. This review will discuss mechanisms of PD-L1 regulation on both tumour cells and tumour-infiltrating immune cells.

Pro-oncogenic pathways, including AKT-mTOR, EGFR, MEK-ERK, and MAPK, are associated with cancer growth and immune evasion and have been linked to PD-L1 expression [10]. For example, the MAPK pathway was activated in melanoma cells resistant to BRAF inhibition through gene mutations and growth factors, leading to increased PD-L1 expression [24]. The PI3K/AKT pathway also plays a role in the upregulation of PD-L1 in cancer cells. Activation of this pathway is caused by mutations of its negative regulators PTEN or SHIP [25,26]. Subsequently, AKT activates the transcription factor, NF-κB, which can upregulate PD-L1 by binding to its promoter and increasing its transcription [7]. STAT3 can also directly increase PD-L1 expression in the same way as NF-κB [27]. All of these regulatory molecules and pathways are possible targets for therapies in cancer, but more research needs to be conducted to reduce toxicity because these pathways are also involved in other processes in the body. 

## 3. Molecules That Regulate PD-L1 Expression 

Some patients do not respond to anti-PD-1/PD-L1 blockades, while for others, the treatment is effective. Also, a patient may initially respond to this therapy but later become resistant. Therefore, it is important to know how PD-L1 is regulated and by what proteins. This information will be important in determining what the resistance mechanisms are and what leads to anti-PD-1/PD-L1 ICBs being ineffective in certain cancer patients. PD-L1 expression is regulated at many levels, including genetic, epigenetic, transcriptional, translational, and posttranslational levels, which is reviewed elsewhere [7,11]. This section of the review will highlight various molecules involved in inflammation and immunity that regulate PD-L1 expression. 

IFNs are highly involved in PD-L1 regulation. IFN-γ, which is the type II IFN, induces immune responses against pathogens through the JAK-STAT pathway [28] (Figure 2). It binds to the IFN-γ receptor (IFNGR), which leads to JAK1 and JAK2 phosphorylation [12]. These JAK proteins then bind to the IFNGR, followed by STAT1, which is phosphorylated by JAK2 [12]. Subsequent translocation of STAT1 to the nucleus induces transcription of IFN stimulated genes (ISGs), including IRF1 and IRF9 [12]. ISGs are involved in cell proliferation, differentiation, apoptosis, and the inflammatory response, among other processes [12]. 

IFN-γ, mainly produced by T cells, is the major inducer of PD-L1 expression, leading to IRF1 binding to the PD-L1 promoter through the JAK-STAT pathway [7,14,29]. IFN-γ can also act through other pathways to increase PD-L1 expression. A study found that through JAK2, IFN-γ triggers activation of the PI3K-AKT pathway and when PI3K was inhibited, PD-L1 levels in lung adenocarcinoma cell lines were reduced [30]. Interestingly, they found that STAT1 was involved in the interaction between the JAK-STAT1 and PI3K-AKT pathways [30]. Other studies have also found that STAT2 and STAT3 are upregulated by IFN-γ, and the PD-L2 promoter is regulated by STAT3 [29]. In comparison, it was found that STAT3 does not mediate PD-L1 expression induced by IFN-γ on tumour cells, which showed the importance of PD-1 being activated through ways other than PD-L1 [7]. IFN-γ can also act within the TME to enhance immune evasion by tumour cells. Tumour-infiltrating T cells expressing IFN-γ upregulate PD-L1 in the TME, specifically on tumour cells and immune cells in the TME, such as tumour-infiltrating macrophages and APCs [31]. This role of IFN-γ shows that even if T cells are able to recognize and reach the TME, which is characteristic of hot tumours, T cell effector functions against the tumour cells are shut down due to increased PD-L1 production by tumour cells and other immune cells in the TME. It has been proposed that an IFN-γ score, calculated from the level of expression of different IFN-γ-induced genes including IRF1, can be used as a prognostic marker to determine the potential for anti-PD-1/PD-L1 therapy for some cancer patients [31]. 

Type I IFNs, including IFN-α, IFN-β, and IFN-ω, are required for inducing immune responses against viral infections [28] (Figure 2). They bind to the IFN-α/β receptor (IFNAR), which consists of the IFNAR1 and IFNAR2 subunits, and signal through JAK1 and TYK2, which phosphorylate the STAT proteins [32]. Phosphorylated STAT1 and STAT2, along with IRF9, form the ISGF3 complex, which binds to the IFN-stimulated response element (ISRE) to control ISGs [32]. Type I IFNs have also been found to be involved in upregulating PD-L1 expression. IFN-α has been shown to induce PD-L1 expression in a human beta cell line and in human islet cells through JAK1, with STAT1 and STAT2 also playing a role [14]. IFN-α also increases PD-L1 expression in DCs through JAK, STAT3, and p38 molecules, which reduces the ability of DCs to stimulate T cells [33]. IFN-β may also play a role in PD-L1 induction. When murine and human lung cancer cells were exposed to IFN-β, IRF9-dependent and -independent pathways were involved in PD-L1 expression [34]. The researchers of this study suggested that in persistent oncovirus infections, such as with hepatitis B virus (HBV) and human papillomavirus (HPV), there may be low IFN-β expression along with IFN-γ expression, which may contribute to increased levels of PD-L1 observed in oncovirus-associated cancers [34]. Therefore, combination therapies may be more useful in treating these types of cancers. The receptor, IFNAR, controls PD-L1 regulation by type I IFNs in myeloid-derived suppressor cells (MDSCs), which are found in the TME [35]. Tumour cells induce PD-L1 expression on MDSCs, which contributes to their support to tumour cells in inhibition of T and NK cells to promote tumour progression [35]. This finding may also support the idea that MDSCs behave like APCs in that they bind to T cells, engaging the PD-1 receptor on T cells through PD-L1 that they express [35]. The researchers also observed IFN-α and IFN-β expression, as well as STAT1 phosphorylation in these MDSCs, suggesting an autocrine mechanism where MDSCs secrete type I IFNs, which bind to IFNAR1 to activate STAT1, thereby upregulating PD-L1 production by these MDSCs [35]. 

Various interleukins (ILs) also play a role in PD-L1 regulation. IL-6 from gliobstoma-conditioned media resulted in increased PD-L1 expression in myeloid cells, where IL-6 was found to activate STAT3 that can then bind to the PD-L1 promoter [36]. IL-6 has been shown to recruit MDSCs and M2 macrophages to the TME, which indicates that targeting IL-6 would reduce immune suppression through decreasing PD-L1 levels as well as preventing tumour-induced assistance from the immune system through MDSCs and M2 macrophages [36]. Additionally, IL-27, produced by macrophages and DCs, induces PD-L1 expression on tumour cells through STAT1 and on peripheral blood monocytes [15]. In epithelial ovarian cancer cells, IL-27 increased PD-L1 expression through STAT1 and STAT3 [37]. It also induced PD-L1 production by monocytes and human prostate and lung cancer cells [37]. Mesenchymal stem cells (MSCs), found in the TME of gastric cancers, express IL-8, which induces PD-L1 expression in gastric cancer cells through STAT3 and mTOR signaling [38]. IL-32g, IL-1α, and IL-10 can also increase PD-L1 expression on peripheral blood monocytes by activating STAT3 or p65 [15]. Diffuse large B-cell lymphoma (DLBCL) cells also produce IL-10 to induce PD-L1 expression on DLBCL cells through the activation of the JAK2/STAT3 pathway [39]. High levels of T_H_17 cells, expressing IL-17 and TNF-α, are found in colon and prostate tumours [40]. Both NF-κB and ERK1/2 signaling are involved in IL-17- and TNF-α-induced PD-L1 production in human colon cancer cells, while only NF-κB signaling is involved in prostate cancer cells [40]. It was also found that IL-17 and TNF-α act individually and not synergistically to upregulate PD-L1 expression in these cancer cells [40]. 

The involvement of a variety of inflammatory and immune proteins/pathways demonstrates the complexity in PD-L1 regulation. Therefore, it is important to monitor during cancer treatments not just PD-L1 upregulation, but also that of these other proteins because they regulate PD-L1 expression, thereby affecting the efficacy of cancer treatments. Additionally, it is likely that these pathways regulate other checkpoint molecules and inhibitory pathways, so simply blocking PD-L1 with an antibody, as is done with anti-PD-L1 ICBs, may not induce T cell activation if other inhibitory molecules are being upregulated. Therefore, it is important to gain a better understanding of the dynamic interactions and kinetics involved in regulating these pathways. Further research on the regulation of these proteins that regulate PD-L1 expression can help us identify potential for side effects when developing new drugs for cancer treatment before the initiation of clinical trials.

## 4. PD-L1 Regulation during Viral Infection 

During an infection, T cells are activated as part of the adaptive immune response, so PD-1 is upregulated to allow the immune system to shut down this T cell response when required [1]. In acute viral infection, the activation of the PD-1/PD-L1 axis controls the level of activity of cytotoxic CD8+ T cells to ensure that it is strong enough to eliminate the virus, but not so toxic that excess inflammation causes tissue damage [1]. Alternatively, in chronic or persistent infections, there is constant TCR stimulation by viral antigens, which leads to upregulation of PD-1 on T cells through NFATc1, a protein that is part of a transcription factor complex that triggers the transcription of PD-1 [41]. Similar to some other co-inhibitory receptors, including CTLA-4, upregulation of PD-1 in a variety of viral infections is correlated with increased number of exhausted T cells [41]. Therefore, along with T cell activation, PD-1 plays a key role in regulating T cell exhaustion since exhausted T cells have been observed to constitutively express PD-1 [41]. Along with PD-1 upregulation, many viral infections also increase the expression of its ligand, PD-L1. This section of the review discusses PD-L1 regulation in acute and chronic viral infections, and provides a rationale for using anti-PD-1/PD-L1 ICBs as a treatment for them.

After entering the host, viruses can regulate PD-L1 expression through a variety of mechanisms. It can be through the recognition of viruses by pattern recognition receptors (PRRs), such as toll-like receptor (TLR)-3, TLR7, TLR8, and TLR9, all of which recognize pathogen-associated molecular patterns (PAMPs) from viruses and bacteria [42]. These PRRs lead to the induction of the type I IFN pathway, which includes IFN-α and IFN-β, two proteins that regulate PD-L1, as mentioned in the previous section of this review [42]. Another way PD-L1 is regulated by viruses is through viral replication, which leads to production of anti-inflammatory cytokines, such as IL-10, that upregulate PD-L1 expression [1]. Viruses can also produce specific viral proteins that induce PD-L1 expression to assist with evasion of the host immune system [1]. With the introduction of foreign material, host immune cells are recruited, which release pro-stimulatory cytokines, such as those mentioned in the previous section of this review, which can lead to the production of PD-L1 to monitor inflammatory processes [3]. Therefore, the purpose of the production of PD-L1 upon viral infection is two-fold: evasion of the immune system by viruses and prevention of excessive host tissue damage. 

Viruses that cause acute viral infection increase PD-L1 levels in immune cells as well as other host cells. For example, avian influenza virus H9N2 subtype increased PD-L1 mRNA and protein levels in pulmonary microvascular endothelial cells and inhibited T cell immune responses and proliferation [43]. Also, researchers observed that murine friend retrovirus (FV)-infected cells which expressed low levels of PD-L1 were efficiently eliminated by cytotoxic T lymphocytes, whereas those with high PD-L1 expression escaped and went on to develop chronic FV infection in mice [44]. In acute LCMV infection, CD169+ macrophages induced PD-L1 expression in the liver through type I IFN response, and absence of these cells resulted in increased viral replication that contributed to T cell exhaustion, severe immunopathology, and death in mice [45]. Viral infection with respiratory syncytial virus (RSV) and Influenza A virus increased PD-L1 expression on macrophages through IFN-β [46]. Most recently, it was found that eosinophils and basophils of patients infected with severe respiratory syndrome coronavirus 2 (SARS-CoV-2) had increased PD-L1 expression and correlated with severity of the Covid-19 disease [47].

As shown with all of the examples of acute viral infections, in the late phase of infection, many immune molecules, including IFN-γ, TNF-α, IL-6, and IL-1β, are released by immune cells to reduce excessive tissue damage by the immune response [1]. However, this tissue damage is not prevented in viral infections that cause viral hemorrhagic fever (VHF). Viruses causing VHF include Ebola virus, hantavirus, Dengue virus, Marburg virus, Lassa fever virus, yellow fever virus, Nairovirus, and rift valley fever virus. Increased PD-L1 and/or PD-1 levels are observed in infections from these viruses, but it is not accompanied by reduction in the severe symptoms of VHF, which include internal bleeding and multi-organ failure [1,9]. For example, hantavirus replication triggers PRRs, like TLR3, which increase PD-L1 expression on DCs and endothelial cells in vitro [48]. In this study by Raftery et al. (2018), while endothelial cells did not induce T cell proliferation, DCs activated CD8+ T cells through CD86, indicating that there is stronger co-stimulation than co-inhibition in these viral infections, leading to symptoms of VHF. Additionally, both, the inflammatory signals from hantavirus and hantaviral nucleocapsid protein, enhanced PD-L1 and PD-L2 expression, leading to the understanding that PD-L1 upregulation by hantavirus is due to IFNs and IFN-independent mechanisms [48].

In persistent infections, while adaptive immune responses attempt to control the viral infection, they also contribute to host tissue damage through those same mechanisms. Exhausted T cells, which are characteristic of persistent infections, are one way in which damage to host tissues is reduced. With more PD-1 expression on CD8+ T cells and PD-1/PD-L1 interaction, these T cells lose more of their effector functions [1,9]. Other co-inhibitory receptors, such as CTLA-4, also contribute to T cell exhaustion [1,9]. Anti-PD-L1 blockade helps to restore some of the effector functions, but this is not a permanent therapy because once it is terminated, the T cells become exhausted again [1,9]. T cell exhaustion does not mean the immune system has stopped fighting the virus; rather the body is preventing excessive inflammation that can be harmful to host tissues [10]. Another way the damage is reduced is through the development of Tregs, which suppress immune responses by effector T cells [49]. However, their protective function can contribute to viral persistence because of reduced antiviral responses. The role of Tregs in viral infection is reviewed elsewhere [50]. 

There are many examples of chronic viral infections upregulating PD-L1. For example, Hepatitis C virus (HCV) encodes the HCV core protein, which induces PD-L1 expression in monocytes through engagement with TLR2 [51]. Hepatocytes infected with HBV release various extracellular vesicles, including virions, subviral particles, and exosomes, all of which are found to increase PD-L1 expression in monocytes [52]. Individuals with HIV express PD-L1 on CD4+ and CD8+ T cells, and antiretroviral therapy resulted in reduced expression of these markers [53]. Also, in this study by Correa-Rocha et al. (2018), soluble PD-L1 levels were increased in the plasma of HIV+ individuals, and expression of membrane-bound PD-L1 on DCs correlated with soluble PD-L1 [53,54]. 

The viruses that cause latent infections, including varicella zoster virus (VZV) and herpes simplex virus (HSV), also upregulate PD-L1 expression. It was found that VZV-infected peripheral blood mononuclear cells (PBMCs), including monocytes, NK cells, NKT cells, B cells, CD4+ and CD8+ T cells, showed increased levels of PD-L1 [55]. Others have shown that HSV-1 infection in the cornea increases PD-L1 expression in the corneal epithlium as well as on NK cells, DCs, neutrophils, and macrophages, which made it difficult to clear the infection [56]. The cornea is an immune privileged site, so PD-L1 is constitutively expressed here—this shows the role of this protein in maintaining balance. Researchers also showed that administration of anti-PD-1/PD-L1 antibodies locally at the cornea increases efficiency of clearing the infection at the site [56]. 

Cancer-causing oncoviruses are also associated with PD-L1 upregulation. For example, Epstein Barr virus (EBV) causes increase in PD-L1 expression on monocytes through TLR signalling and increased reactive oxygen species [57]. HPV-infected oropharyngeal squamous cell cancer cells were also found to be more likely to express PD-L1 [58]. A study found that infection of monocytes with Kaposi’s Sarcoma-associated Herpesvirus (KSHV) led to increased PD-L1 expression in human monocytes [59]. KSHV is mostly seen in immunocompromised individuals, including HIV-positive patients. Many individuals with acquired immunodeficiency syndrome (AIDS) have monocytes co-infected with KSHV along with HIV, so PD-L1 upregulation allows for viral evasion as well as oncogenic progression [59]. Along with being an oncovirus, KSHV is also a persistent and latent infection. Researchers suggested that chronically increased PD-L1 levels on monocytes may result due to repeated infection after reactivation of KSHV [59]. 

Increased PD-L1 expression seen in the vast number of viruses that cause acute, chronic/persistent, and/or latent infections, as well as those that cause cancer, leads to the speculation of whether anti-PD-1/PD-L1 ICBs may be a possible therapy for viral infections. Indeed, anti-PD-1/PD-L1 ICBs have been tested to be used to treat patients with different viral infections in order to activate T cells or reactivate exhausted T cells [60]. These therapies need to be used carefully because excessive blocking of the PD-1/PD-L1 axis can result in autoimmune conditions, which is one of the side effects of anti-PD-1/PD-L1 ICBs [60]. A solution to this approach is to use combination therapies, as used with cancer treatments, to reduce or eliminate viral infection, while preventing the development of autoimmune conditions. Researchers treated woodchucks infected with woodchuck hepatitis virus (WHV) (comparable to HBV in humans) who had elevated PD-L1 levels with anti-PD-L1 monoclonal antibody therapy along with entecavir, a drug to treat HBV [61]. This combination therapy showed improved control of viraemia compared to entecavir treatment alone in some, but not all, animals. Other researchers have shown that combining anti-PD-L1 blockade with OX40 stimulation can also improve CD4+ T cell responses against HBV by increasing IFN-γ and IL-21 secretion from helper T cells [62]. A clinical study used PD-L1 blockade on patients with HIV-1 infection who were on antiretroviral therapy and found that there were improvements in HIV-1-specific CD8+ responses, although some patients showed toxicity with higher doses of ICB [63]. Another study showed that combining therapeutic vaccination with anti-PD-L1 antibody to treat Friend retrovirus improved CD8+ T cell responses and viral clearance, while promoting reactivation of exhausted CD8+ T cells [64]. The results from all of these combination therapies using anti-PD-1/PD-L1 ICBs to treat viral infection show that there is potential for this therapy, but more research needs to be conducted to determine the effectiveness and dosage levels of anti-PD-1/PD-L1 therapeutics in the context of viral infections. 

## 5. Modulation of PD-L1 by Cancer Therapies 

### 5.1. Anti-PD-1/PD-L1 ICBs

The anti-PD-1/PD-L1 ICBs pembrolizumab and nivolumab target PD-1, while atezolizumab, avelumab, and durvalumab target PD-L1. These drugs are either approved or in clinical trials for a variety of cancers, including melanoma, lung cancer, urothelial carcinoma, head and neck squamous cell carcinoma, renal cell carcinoma, cutaneous squamous cell carcinoma, triple negative breast cancer, and Merkel cell carcinoma [65]. The mechanisms through which anti-PD-1/PD-L1 ICBs treat cancer cells include increasing tumour-infiltrating T cells that produce inflammatory mediators, such as IFNs, and inducing cytotoxicity of CD8+ T cells to kill tumour cells [10]. Despite the understanding of the mechanisms of anti-PD-1/PD-L1 ICBs, there is often reduced efficacy of these anti-cancer drugs in cancer patients. 

One reason for the reduced efficiacy is innate resistance, where individuals may have tumours that lack T cells, so anti-PD-1/PD-L1 ICBs are ineffective, since their target (i.e., T cells) is absent from the TME. Lack of T cells in the TME may be due to a variety of tumour-intrinsic mechanisms, which are reviewed elsewhere [66,67,68]. Another reason is acquired resistance, where patients may respond to ICBs initially, but the therapy may later have reduced efficacy or be completely ineffective. This can be due to mutations of neoantigens or proteins that are part of immunostimulatory pathways [66,67,68]. Age, genetics, epigenetics, dysbiosis, diet and metabolism can also affect the efficacy of ICBs [66,68]. 

Other factors outside of the TME have also been shown to influence the response to anti-PD-1/PD-L1 therapy. There is emerging evidence suggesting that gut microbiota can play an important role in determining the response to ICB therapy in patients with melanoma, NSCLC, renal cell carcinoma and GI cancer [69,70,71,72]. One such study published in *Science* reports that the use of antibiotics in patients with advanced cancer diminished their response to anti-PD-1/PD-L1 therapy. Furthermore, the authors demonstrated that Fecal microbiota transplantation (FMT) from responsive cancer patients into antibiotic-treated mice enhanced the antitumour effects of PD-1 blockade, while FMT from non-responding patients failed to do so [71]. The same study identified a relative abundance of *Akkermansia muciniphila* to be responsible for improving the response to PD-1 blockade. A similar study by Peng and colleagues assessed the role of gut microbiota in patients with advanced GI cancer by collecting fecal samples prior to and during anti-PD-1/PD-L1 therapy. They found that gut bacteria capable of producing short-chain fatty acids (SCFA), including Eubacterium, Lactobacillus, and Streptococcus, were positively associated with anti–PD-1/PD-L1 response across different GI cancer types [72]. The molecular mechanisms by which microbiota can regulate response to ICB therapy remains unclear, however several studies have shown that microbiota can influence antitumour immunity and the production of IFN-γ-producing CD8+ T cells [73,74,75]. The strong association between gut microbiota and response to ICB therapy will enable the prospect of using the microbiome as a biomarker for response to ICB therapy. Furthermore, as the molecular mechanisms responsible for this interaction are uncovered, a greater emphasis can be placed on modulating the gut microbiota to sensitize otherwise non-responding patients to ICB therapy.

Besides resistance, clinical trials have also shown some level of toxicity with the use of anti-PD-1/PD-L1 ICBs. Some severe side effects (referred to as immune-related adverse events) include development of autoimmune conditions (including hyperthyroidism, type I diabetes, and autoimmune adrenalitis) and conditions like hepatitis, nephritis, and hypophysitis [65,76]. Many of the illnesses are autoimmune because the T cells are not recognizing self antigens as self [76]. 

Using anti-PD-1/PD-L1 ICBs in conjunction with other cancer therapies will likely be more effective, since the therapy can act to kill the tumour cells, while the ICB can promote the activation of the host immune system so it can act with the therapy to eliminate the tumour. Combination therapy may also reduce the toxicity of anti-PD-1/PD-L1 ICBs and reduce the likelihood of acquired resistance. Latest research on various combination therapies with anti-PD-1/PD-L1 ICBs will be discussed later in the review.

### 5.2. Other Cancer Therapies 

Chemotherapy is the most-used therapy for a variety of cancer types. Many studies have shown increased PD-L1 levels after treatment with chemotherapeutic drugs. For example, in patients with thymic epithelial tumours who were treated with chemotherapy, tumour cells expressed increased PD-L1 and tumour-infiltrating immune cells expressed increased PD-1 [77]. Chemotherapeutic drugs also led to increased levels of PD-1 and PD-L1 in NK cells and nasopharyngeal carcinoma cells, respectively, through NF-κB [78]. Researchers found that adjuvant chemotherapy may upregulate PD-L1 in patients with NSCLC [79]. Cervical cancer patients who had neoadjuvant chemotherapy had higher PD-L1 expression compared to those without neoadjuvant chemotherapy [80]. The same result was seen with epithelial ovarian cancer [81]. In esophageal squamous cell carcinoma, PD-L1 expression is increased after chemotherapy via the EGFR/ERK pathway [82]. Other studies have shown that the MAPK pathway may upregulate PD-L1 in cancer cells treated with chemotherapeutic drugs [7,11]. These results show that the pathway through which PD-L1 is induced after chemotherapy depends on cancer type. 

As already shown, many studies have observed PD-L1 and/or PD-1 upregulation with chemotherapy. However, some studies have also reported downregulation. For example, White et al. (2021) showed that some patients with malignant peritoneal mesothelioma had downregulation of PD-L1 after chemotherapy, while others had upregulation [83]. Rojkó et al. (2018) reported similar results in lung cancer patients [84]. This heterogeneity in PD-L1 expression with chemotherapy may be attributed to the type of cancer and chemotherapeutic drug, both of which should be considered when combining chemotherapy and ICBs for treatment.

Targeted therapies have also been used to treat specific types of cancer and have shown to upregulate PD-L1. For example, one study found that poly ADP ribose polymerase (PARP) inhibitor, which is used for ovarian and breast cancer, upregulates PDL-1 expression through GSK3β inactivation [85]. Another study showed that histone deacetylase 3 (HDAC3) inhibitor upregulates PD-L1 expression through epigenetic mechanisms on DCs in the TME in B-cell lymphomas [86]. Researchers have also shown that clear cell renal cell carcinoma (ccRCC) treated with VEGFR-tyrosine kinase inhibitors (TKIs) upregulates PD-L1 and PD-1 expression on immune cells in the TME [87]. There is also evidence that continuous treatment with EGFR-TKIs can lead to upregulation of PD-L1 in resistant NSCLC [88]. However, another study found that EGFR-TKIs can downregulate PD-L1 expression in EGFR-mutant NSCLC through NF-κB signaling [89]. Therefore, it is important to test different targeted therapies on different types of cancer when considering combination therapy with anti-PD-1/PD-L1 ICBs. 

Radiation therapy is one of the oldest therapies for cancer and is still used today. Many studies have shown that fractionated radiotherapy results in increased PD-L1 levels. For example, studies have found PD-L1 upregulation in melanoma cells and head and neck squamous cell carcinoma tumour cells after fractionated radiotherapy [90,91]. Similar results have been reported in certain lung cancer cell lines [92]. Irradiation of murine and human bladder cancer cells resulted in increased PD-L1 expression in vitro and dosage of irradiation correlated with PD-L1 expression in vivo in murine models [93]. One study found that radiation therapy upregulated PD-L1 expression in NSCLC, which may be through the PI3K/AKT and STAT3 pathways [94]. Fractionated radiotherapy combined with chemotherapy (radiochemotherapy) increased PD-L1 expression on glioblastoma cells, but has no significant impact on PD-L1 expression in colorectal cancer cells, showing that efficacy of combination therapy with anti-PD1/PD-L1 ICBs and radiochemotherapy depends on cancer type and chemotherapeutic drug type [91]. 

Vaccination is another therapy for cancer, which is still being researched. Some studies have shown that this therapy increases PD-L1 levels. For example, immunization in murine models and in humans with a DNA vaccine encoding the prostate tumour antigen, synovial sarcoma X breakpoint 2 (SSX2), resulted in increased PD-L1 expression on tumour cells via IFN-γ secretion by T cells [95,96]. Researchers have also observed increased PD-L1 expression in pancreatic ductal adenocarcinoma tumours from patients treated with a pancreatic cancer vaccine that secretes granulocyte-macrophage colony-stimulating factor (GM-CSF) [97]. These results show that cancer immunization may be a potential candidate for combination therapy with anti-PD-1/PD-L1 ICBs for which more research is required. 

As highlighted in the previous section of this review, viruses are highly capable of inducing inflammatory responses, which leads to increased PD-L1 expression. This property of viruses has been used to develop oncolytic virotherapy as another cancer immunotherapy. Oncolytic viruses are viruses that kill cancer cells directly through oncolysis. They can also do so indirectly by redirecting the host immune system to recognize tumour cells, inducing an immunostimulatory response through upregulation of MHCs, and releasing PAMPs, damage-associated molecular patterns (DAMPs), and other inflammatory cytokines and chemokines [98,99]. Oncolytic viruses can also target tumour vasculature to cut off blood and oxygen supply preventing tumour growth and survival [98,99]. This therapy has proven very useful, especially in cold tumours, where oncolytic viruses are able to create an immunogeneic environment to attract the host immune cells [6]. However, this therapy has also shown that it leads to PD-L1 upregulation. One study showed that oncolytic vaccinia virus upregulates PD-L1 expression in tumour cells and immune cells in the TME [100]. Using nonpathogenic Newcastle disease virus (NDV) as an oncolytic virus resulted in increased infiltration of effecter T cells, but that was eventually inhibited through upregulation of PD-L1 in tumour cells and immune cells in the TME [101]. Researchers found that using Semliki Forest virus encoding IL-12 (SFV-IL12) as an oncolytic virus also resulted in increased PD-L1 expression via IFN-γ in murine colon cancer and melanoma cells [102]. These results show that oncolytic viruses can be combined with anti-PD-1/PD-L1 ICBs to get better treatment outcomes in cancer patients.

### 5.3. Combination Therapies with anti-PD-1/PD-L1 ICBs 

Overall, researchers have shown that chemotherapy, certain targeted therapies, radiotherapy, vaccination, and oncolytic viruses upregulate PD-L1 in certain cancers, which makes them ideal candidates to be tested for combination therapy with anti-PD-1/PD-L1 ICBs. In this section of the review, we discuss the results from a number of studies that combined anti-PD-1/PD-L1 ICBs with different cancer therapies and showed that combination therapy is more efficacious than the two therapies alone (Table 1).

Oncolytic viruses are often combined with anti-PD-1/PD-L1 ICBs, since the former result in immune responses that lead to PD-L1 upregulation. Thus, PD-1/PD-L1 ICBs are used to improve their efficacy while also preventing T cell inhibition [104]. In mouse models with ovarian cancer or colon cancer, combination of oncolytic vaccinia virus and anti-PD-L1 treatment lowered Tregs, exhausted CD8+ T cells, and PD-L1 expression in the TME, while increasing effector T cells [100]. Combining NDV with anti-PD-1/PD-L1 ICB improved efficacy compared to using NDV alone [101]. Researchers also observed that combining SFV-IL12 with ICBs resulted in regression of the tumour and increased survival of murine models [102]. Using oncolytic vesicular stomatitis virus that encodes murine IFN-β and the sodium iodide symporter (VSV-mIFNB-NIS) with anti-PD-L1 antibody in mice that had acute myeloid leukemia resulted in improved host anti-tumour response, increased survival of the mice, and increase in tumour-infiltrating T cells [103]. It is important to note that VSV-mIFNB-NIS treatment alone did not increase PD-L1 on tumour cells, but the combination with anti-PD-L1 antibody showed improvement, which indicates that upregulation of PD-L1 by the cancer therapy is not a necessary prerequisite of the combination of that cancer therapy with anti-PD-1/PD-L1 ICBs being successful. Indeed, many studies have reported increased efficacy in generating an immune response against a variety of tumours using anti-PD-1/PD-L1 ICBs with different oncolytic viruses, including various DNA and RNA viruses, which are reviewed elsewhere [104]. 

Targeted therapies for cancer have also been tested for their efficacy as combination therapies with anti-PD-1/PD-L1 ICBs. One study reported that using IFN-α therapy with anti-PD-1/PD-L1 blockades may be effective at promoting the immunostimulatory effects of IFN-α, including DC activation and subsequent T cell activation [33]. It has been seen with neuroblastoma that viral infection promotes immunogenicity of tumour cells—this finding led to the study by Boes and Meyer-Wentrup (2015), who showed that triggering TLR3 promotes PD-L1 upregulation and IL-8 secretion and that this can be applied to combination therapies using ICBs and synthetic TLR ligands [105]. Combined therapy with VEGF-TKI and anti-PD-L1/PD-1 ICB have also shown significant improvement in patients with hepatocellular carcinoma [106]. Combining HDAC3 inhibitor with anti-PD-L1 immunotherapy resulted in tumour regression in a murine lymphoma model [86]. Researchers saw that blocking PD-L1 in cancer cells treated with PARP inhibitor allowed the cancer cells to be killed by T cells again [85]. This study provide a rationale to test combination therapy using ICBs and PARP inhibitor in patients with ovarian and/or breast cancer.

Other combination therapies with anti-PD-1/PD-L1 ICBs have also shown potential to be used as cancer treatments. A phase-3 clinical trial of first-line treatment with combination therapy consisting of the chemotherapy drug, nanoparticle albumin-bound (nab)-paclitaxel, and anti-PD-1/PD-L1 antibody in patients with triple-negative breast cancer was conducted [107]. The researchers found that patients treated with this combination therapy had prolonged progression-free survival compared to those treated with placebo and nab-paclitaxel. In a systematic review and meta-analysis, Dafni et al. (2019) showed that combination of chemotherapy with anti-PD-1/PD-L1 ICBs enhanced treatment efficacy as a first-line treatment in NSCLC patients compared to using chemotherapy alone [108]. Combining anti-cancer vaccines with ICBs for prostate cancer showed antigen-specific immunity [95,96]. Similar results were seen for pancreatic ductal adenocarcinoma in murine models—it resulted in increased effector CD8+ T cells in the TME and reduced Tregs [97]. Combining radiation therapy and anti-PD-L1 antibody increased CD8+ T cells in the TME, while reducing MDSCs and Tregs in an NSCLC mouse model [94]. Indeed, many studies have reported increased efficacy in treatment combining radiation therapy and anti-PD-1/PD-L1 ICBs, with some in different stages of clinical trials [109]. Anti-PD-1 ICB improves the efficacy of chimeric antigen receptor (CAR) T cells and can rescue exhausted CAR T cells [110]. Of interest, some studies use engineered CAR T cells that have the ability to block the PD-1/PD-L1 pathway, making it unnecessary to use anti-PD-1/PD-L1 ICBs in combination with them [110,111,112]. These studies have shown efficacy of the modified CAR T cells in gastric cancer, pleural mesothelioma, and renal cell carcinoma cells. Calagua et al. (2018) found that administering hormone therapy in prostate cancer patients did not increase PD-L1 expression, which shows that not all cancer therapies lead to PD-L1 upregulation [113]. Therefore, it is important to research combination of anti-PD-1/PD-L1 ICBs with every drug in every type of cancer, because levels of efficacy and toxicity may be dependent on the drug and/or cancer.

## 6. Conclusions

It is important to note that there are many redundant processes in the human body and the immune system is no different. If the PD-1/PD-L1 axis is targeted in therapeutics, there are other molecules or pathways that cancer cells or viruses can use to evade the immune system, which is why we see resistance to some drugs. This idea further gives rationale for combination therapies for cancer. Most combination therapies with anti-PD-1/PD-L1 ICBs not only show greater efficacy compared to anti-PD-1/PD-L1 ICBs alone, but also compared to the other therapy when it is used alone. This result is seen with oncolytic viruses, some chemotherapeutic drugs, vaccination, some targeted therapies, oncolytic viruses, and radiation therapy. More research is required testing the combination of various existing cancer therapies with anti-PD-1/PD-L1 ICBs on different cancers, so toxicity and efficacy can be determined before they are used in patients. 

## Figures and Tables

**Figure 1 ijms-22-04893-f001:**
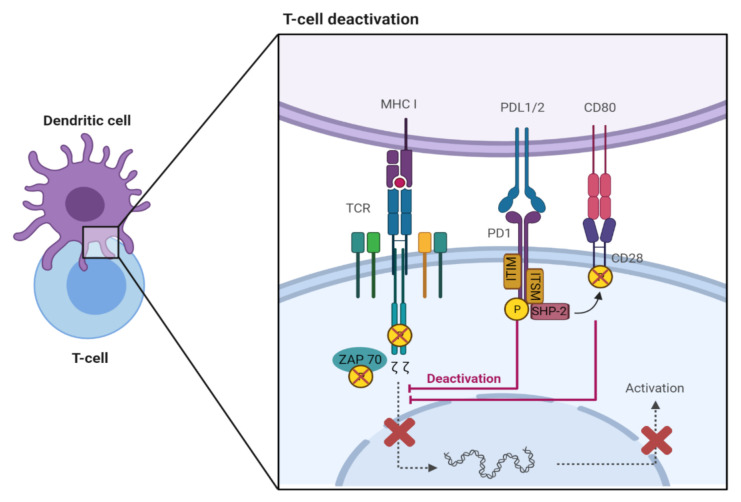
Downstream effects of PD-1/PD-L1 ligation, leading to inhibition of T cell activation. T cell recognition of foreign antigens presented by MHC on the surface of antigen-presenting cells (APCs), or tumour cells initiates downstream T cell receptor (TCR) signaling, leading to the activation and differentiation of T cells. This process is regulated by co-stimulatory and inhibitory interactions, such as CD80/CD28 and PD-L1/PD-1 interactions, respectively. Binding of PD-1 to its ligand results in the dephosphorylation of Zeta-chain-associated protein kinase 70 (ZAP70) and inhibition of downstream TCR signaling.

**Figure 2 ijms-22-04893-f002:**
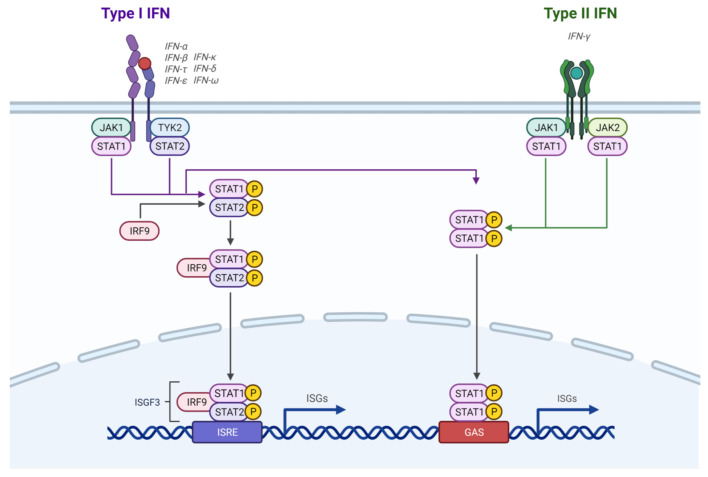
Type I and Type II interferon signaling pathways. Upon engagement of their respective ligands, the interferon-α receptor (IFNAR) and interferon-γ receptor (IFNGR) complexes activate downstream kinases. The kinases, in turn, phosphorylate signal transducer and activator of transcription (STAT) proteins resulting in their dimerization and nuclear translocation. Engagement of the IFNAR receptor complex results in the formation of the interferon-stimulated gene factor 3 (ISGF3) complex (composed of STAT1, STAT2, and IFN-regulatory factor 9 (IRF9)) which binds to IFN-stimulated response element (ISRE) sequence to initiate transcription of antiviral genes. Engagement of the IFNGR complex results in the phosphorylation, homodimerization and nuclear translocation of STAT1, where the homodimers bind to IFN-γ-activated sequence (GAS) elements and initiate the transcription of inflammatory genes.

**Table 1 ijms-22-04893-t001:** Cancer therapies combined with anti-PD-1/PD-L1 immune checkpoint blockades (ICBs). IFN = interferon, TLR = toll-like receptor, TKIs = tyrosine kinase inhibitors, HDAC3 = histone deacetylase 3, PARP = poly ADP ribose polymerase, NSCLC = non-small cell lung cancer, CAR = chimeric antigen receptor.

Therapy Combined with Anti-PD-1/PD-L1 ICBs	Examples
**Oncolytic viruses**	1. Using oncolytic vaccinia virus and anti-PD-L1 ICBs in murine ovarian and colon cancer cells increased CD4+ and CD8+ tumour-infiltrating T cells, while reducing PD-L1-expressing immune cells in the tumour microenvironment [100]. 2. Combining Newcastle disease virus and anti-PD-1/PD-L1 ICBs in colon adenocarcinoma and melanoma murine models resulted in T cell infiltration and rejection of tumours [101]. 3. Treating melanoma, colon cancer, and breast cancer murine models with a combination of Semliki Forest virus encoding IL-12 (SFV-IL12) and anti-PD-1 ICB improved survival [102]. 4. Using anti-PD-L1 ICB and oncolytic vesicular stomatitis virus encoding murine IFN-β and the sodium iodide symporter (VSV-mIFNB-NIS) in an acute myeloid leukemia murine model increased tumour-infiltrating and tumour-specific T cells [103]. 5. Many combination therapies are in clinical trials that combine anti-PD-1/PD-L1 ICBs with other RNA or DNA oncolytic viruses, such as herpes simplex virus, adenovirus, myxoma virus, reovirus, maraba virus, and measles virus [104].
**IFN-α therapy**	IFN-α increased PD-L1 expression in mouse leukocytes and human dendritic cells, suggesting that combining IFN-α therapy with anti-PD-L1 ICBs could improve treatment efficacy [33].
**TLR ligands**	PD-L1 expression was induced in neuroblastoma cells when TLR3 was triggered, suggesting that a combination of synthetic TLR3 ligands and anti-PD-L1 ICB may be a potential therapy for neuroblastoma [105].
**TKIs**	In a phase Ib clinical trial with hepatocellular carcinoma patients, combining VEGF-TKI and anti-PD-L1 ICB improved patient response rates and survival [106].
**HDAC3 inhibitor**	Combining an HDAC3 inhibitor with anti-PD-L1 ICB increased tumour regression, compared with each therapy alone, in a murine lymphoma model [86].
**PARP inhibitor**	In breast cancer murine models, using PARP inhibitor with anti-PD-L1 ICB resulted in tumour growth restriction and restoration of tumour-infiltrating CD8+ T cells [85].
**Chemotherapy**	1. In a phase III clinical trial, a combination of nanoparticle albumin-bound (nab)-paclitaxel and anti-PD-L1 ICB in triple-negative breast cancer patients increased progression-free survival [107]. 2. In a meta-analysis, other chemotherapeutic drugs combined with ICBs, including those of anti-PD-1 or anti-PD-L1, showed improved overall survival and progression-free survival in patietns with NSCLC [108].
**Anti-cancer vaccines**	1. Mice treated with anti-PD-1/PD-L1 ICB and a DNA vaccine encoding the prostate tumour antigen, synovial sarcoma X breakpoint 2 (SSX2), had increased CD8+ tumour-infiltrating lymphocytes [95]. When peripheral blood mononuclear cells, from prostate cancer patients who were given a DNA anti-cancer vaccine, were treated with anti-PD-1 ICB, prostate antigen-specifc immunity was observed [96]. 2. Combining anti-PD-1 ICB with a pancreatic cancer vaccine that secretes granulocyte macrophage colony-stimulating factor resulted in improved survival of murine models and increased CD8+ T cells in the tumour microenvironment [97].
**Radiation therapy**	1. Combining radiation therapy with anti-PD-L1 ICB in an NSCLC murine model promoted infiltration of CD8+ T cells and reduced myeloid derived suppressor cells and T regulatory cells [94]. 2. Many combination therapies with radiation therapy and anti-PD-1/PD-L1 ICBs are in pre-clinical and clinical trials for different types of cancer, including melanoma, triple-negative breast cancer, myeloma, colon cancer, renal cancer, head and neck squamous cell cancer, bladder cancer, lung cancer, glioma, and pancreatic cancer [109].
**CAR T cells**	1. In a mouse model of pleural mesothelioma, using CAR T cells with CD28 and anti-PD-1 ICB reduced tumour burden [110]. Genetically engineered CAR T cells that intrinsically resist PD-1 also reduced tumour burden and prolonged survival [110]. 2. Treating gastric cells with a bispecific CAR T cell targeting PD-L1 and a gastric cancer antigen resulted in the CAR T cells expressing IFN-γ and killing tumour cells [111]. 3. CAR T cells engineered to secrete PD-L1 antibodies resulted in reduced renal cell carcinoma tumour growth in murine models, reduced T cell exhaustion, and recruited NK cells to the tumour [112].

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
