# Peer review of "Mechanisms of PD-L1 Regulation in Malignant and Virus-Infected Cells"

_ijms, 2021, doi:10.3390/ijms22094893_

Round 1

Reviewer 1 Report

Manuscript ID: ijms-1180382

A timely review on PD-1/PD-L1 regulation summarizes latest knowledge in the field and extends to combination therapies with anti-PD-1/PD-L1 antibodies.

Although many papers have described overexpression of PD-L1 in cancer specimen, most did not discriminate between overexpression in tumour cells and overexpression in immune cells infiltrating the tumour. The authors should discuss this important point and indicate in which papers PD-L1 were found to be upregulated in the tumour cells.

Authors may also discuss some preclinical and clinical studies suggesting microbiota have played important roles in determining the response to ICB.

Minor:

Figure 1 and 2. Define some abbreviations. The caption could be more informative.

Author Response

Reviewer 1:

A timely review on PD-1/PD-L1 regulation summarizes latest knowledge in the field and extends to combination therapies with anti-PD-1/PD-L1 antibodies. 

Although many papers have described overexpression of PD-L1 in cancer specimen, most did not discriminate between overexpression in tumour cells and overexpression in immune cells infiltrating the tumour. The authors should discuss this important point and indicate in which papers PD-L1 were found to be upregulated in the tumour cells.

Author’s response: We thank the reviewer for their positive feedback and helpful suggestions. We added a section in which we discuss the effects of PD-L1 expression on both tumour cells and immune cells in the TME, and expanded on the clinical differences observed depending on what cell types upregulate PD-L1 (lines 147-159). Furthermore, we added more clarification throughout the review and included the cell types that were shown to express PD-L1 when reviewing papers (lines 218, 270, 280, 283, 393, and 403).

Authors may also discuss some preclinical and clinical studies suggesting microbiota have played important roles in determining the response to ICB.

Author’s response: We thank this reviewer for their suggestion. We believe it is a valuable addition that is highly relevant to the topic. We added a section in which we discuss how microbiota can influence the response to ICB therapy in both clinical and pre-clinical settings and highlighted how this area of research has important implications for the future of immunotherapy (lines 456-477).

Figure 1 and 2. Define some abbreviations. The caption could be more informative.

Author’s response: We added more detailed captions to both figures and defined abbreviations as requested.

Reviewer 2 Report

In the present review, the authors discuss the mechanisms of PD-L1 regulation in malignant and virus-infected cells. The manuscript is well written and provides latest information on molecules of inflammation and innate immunity that regulate PD-L1 expression, its regulation during viral infection, and how it is modulated by different cancer therapies. Such knowledge might help to improve the tumor response to immunotherapy by modulating the tumor microenvironment and decreasing the immunosuppression. I have no further suggestions.

Author Response

Reviewer 2:

In the present review, the authors discuss the mechanisms of PD-L1 regulation in malignant and virus-infected cells. The manuscript is well written and provides latest information on molecules of inflammation and innate immunity that regulate PD-L1 expression, its regulation during viral infection, and how it is modulated by different cancer therapies. Such knowledge might help to improve the tumor response to immunotherapy by modulating the tumor microenvironment and decreasing the immunosuppression. I have no further suggestions.

Author’s response: We thank the reviewer for their positive feedback.